# RELMα Is Induced in Airway Epithelial Cells by Oncostatin M without Requirement of STAT6 or IL-6 in Mouse Lungs In Vivo

**DOI:** 10.3390/cells9061338

**Published:** 2020-05-27

**Authors:** Lilian Ho, Ashley Yip, Francis Lao, Fernando Botelho, Carl D. Richards

**Affiliations:** McMaster Immunology Research Centre, Department of Pathology and Molecular Medicine, McMaster University, Hamilton, ON L8P, Canada; hol2@mcmaster.ca (L.H.); yipa1@mcmaster.ca (A.Y.); laof@mcmaster.ca (F.L.); botelhf@mcmaster.ca (F.B.)

**Keywords:** Oncostatin M, RELMα, airway epithelial cells, lung inflammation

## Abstract

Resistin-like molecule alpha (RELMα) and YM-1 are secreted proteins implicated in murine models of alternatively activated macrophage (AA/M2) accumulation and Th2-skewed inflammation. Since the gp130 cytokine Oncostatin M (OSM) induces a Th2-like cytokine and AA/M2 skewed inflammation in mouse lung, we here investigated regulation of RELMα and YM-1. Transient pulmonary overexpression of OSM by Adenovirus vector (AdOSM) markedly induced RELMα and YM-1 protein expression in total lung. In situ hybridization showed that RELMα mRNA was highly induced in airway epithelial cells (AEC) and was co-expressed with CD68 mRNA in some but not all CD68+ cells in parenchyma. IL-6 overexpression (a comparator gp130 cytokine) induced RELMα, but at significantly lower levels. IL-6 (assessing IL-6^−/−^ mice) was not required, nor was STAT6 (IL-4/13 canonical signalling) for AdOSM-induction of RELMα in AEC. AEC responded directly to OSM in vitro as assessed by pSTAT3 activation. RELMα-deficient mice showed similar inflammatory cell infiltration and cytokine responses to wt in response to AdOSM, but showed less accumulation of CD206+ AA/M2 macrophages, reduced induction of extracellular matrix gene mRNAs for COL1A1, COL3A1, MMP13, and TIMP1, and reduced parenchymal alpha smooth muscle actin. Thus, RELMα is regulated by OSM in AEC and contributes to extracellular matrix remodelling in mouse lung.

## 1. Introduction

Chronic respiratory diseases, such as idiopathic pulmonary fibrosis (IPF), chronic obstructive pulmonary disease (COPD), and severe asthma, are conditions that imply a reduction in lung function that involves extracellular matrix (ECM) remodelling [1]. In COPD, damaged alveolar walls and the loss of shape and structure in larger airways reduce gas exchange efficiency [2], whereas IPF and severe asthma are characterized by excessive ECM deposition in the airways or around alveoli, therefore decreasing gas exchange with the capillary network and consequently impairing pulmonary function [1,3,4]. These conditions are often progressive in nature and current therapies have much room for improvement [3,5]. Animal models of lung fibrosis implicate activated T cells and alternatively activated macrophages in the induction of ECM [6,7]. Although cytokines such as transforming growth factor beta (TGFβ) [8] and interleukin-4 (IL-4) [9] are critical factors in fibrogenesis in many systems, studies have also shown pathways independent of TGFβ/SMAD3 and IL-4/STAT6 that can lead to fibrosis in animal models in vivo [10]. Thus, understanding the various mechanisms of how ECM remodelling occurs and through which cells and signaling molecules will provide a better understanding to inform new potential therapeutic approaches separate from the TGFβ pathway.

Several members of the gp130 cytokine family have been implicated in inflammatory responses of chronic respiratory diseases and ECM remodelling in the lung, including Oncostatin M (OSM) and IL-6 [11,12,13,14]. OSMRβ and gp130 receptor chains, required for the active OSM receptor complex on cell surfaces [12], are widely expressed in stromal/connective tissue cells, including those responsible for ECM remodelling in the lung [15,16]. Although receptors for other gp130 cytokines, such as IL-6Rα, may also be expressed on connective tissue cells, their expression are relatively lower than OSMRβ [17]. In addition, we have previously shown that OSM is able to activate such cells more robustly than other cytokines in this family [13,18]. Furthermore, OSM has been detected at elevated levels in the broncho-alveolar lavage (BAL) fluid of IPF patients, sputum of asthma and COPD patients, as well as in allergic rhinitis tissue, suggesting that OSM may be participating in the pathogenesis of these chronic diseases [11,19,20,21]. We have previously shown that overexpression of OSM induces ECM accumulation and accumulation of arginase-1^+^ (Arg1) alternatively activated (AA/M2-like) macrophages, which was associated with eosinophilia and a Th2-like skewed cytokine profile in lungs of C57Bl/6 mice [22]. In preliminary work for the present study, we observed a marked induction of the gene for resistin-like molecule alpha (RELMα), a typical AA/M2 macrophage product. Regulation of RELMα by OSM has not been previously described, and whether RELMα participates in OSM-induced inflammation and pathology is examined here.

RELMα, also known as found in inflammatory zone 1 (FIZZ1), is a member of the RELM/FIZZ family of cysteine-rich secreted proteins that share a highly conserved signature 10-cysteine residues motif in the C-terminal domain [23,24]. RELMα (9.4 kDa) can be secreted by several different cell types including AA/M2 macrophages, eosinophils, B lymphocytes, adipocytes, and alveolar epithelial cells [23,25,26,27,28]. In AA/M2 macrophages, RELMα can be up-regulated by IL-4 or IL-13 in vitro, and its dependence on STAT6 in such cells was confirmed using STAT6 knockout mice [29,30]. RELMα has been implicated in murine models of experimental asthma, parasitic infection, pulmonary fibrosis, and wound healing [23,26,31,32,33,34,35]. Liu et al. showed that RELMα-deficient mice challenged with bleomycin were protected from the fibrotic phenotype, suggesting a profibrotic role for RELMα [33]. In a different model, Nair et al. demonstrated that RELMα-deficient mice challenged with parasitic infection developed exacerbated lung inflammation associated with elevated Th2 cytokine expression, presenting a healing phenotype [26]. Given the proposed functions of RELMα in mouse lung inflammation, in this study we evaluate the regulation of RELMα by gp130 cytokines OSM and IL-6 in vivo and examine the roles of RELMα in OSM-induced lung inflammation and ECM gene expression in vivo using RELMα-deficient mice. We show here that OSM is a potent inducer of RELMα in airway epithelial cells, markedly increases RELMα levels locally in lung, and that RELMα contributes to the ECM remodelling processes induced by OSM.

## 2. Materials and Methods

### 2.1. Mice and Cell Culture

Female BALB/c wild-type (6–8 weeks old) were purchased from Charles River Laboratories (Wilmington, MA, USA). Female C57Bl/6 wild-type (6–10 weeks old), IL-6^–/–^ (C57Bl/6 background; 6–8 weeks old), and RELMα^–/–^ mice (C57Bl/6 background; 8–10 weeks old) purchased from The Jackson Laboratory (Bar Harbor, ME, USA) were acclimatized one week prior to experimental procedures, and housed under specific pathogen-free conditions within the McMaster University Central Animal Facility. All experimental procedures were approved by the McMaster University Animal Research Ethics Board. Mice were endo-tracheally administered with sterile PBS or 5 × 10^7^ PFU of AdDel70, AdOSM, or 3 × 10^7^ PFU of AdIL-6. Animals were culled 2, 7, or 14 days following administration.

### 2.2. Airway Epithelial Cells

C57Bl/6 mouse adult Tracheal-Bronchial Epithelial Cells (TBE cells; Creative Bioarray (cat# CSC-9087J), Shirley, NY, USA) were cultured in SuperCult^®^ (Creative Bioarray). Complete Mouse Epithelial Cell culture basal medium (containing 0.1% insulin-transferin-selenium (ITS), 0.1% epidermal growth factor (EGF), 1% L-glutamine, 1% antibiotic-antimycotic solution and 2% fetal bovine serum). Cells were grown on sterile culture flasks coated with a 1% gelatin solution (Cell Attachment 1 × (cat#S006100) from ThermoFisher Scientific). For cell stimulation, cells were stimulated in complete mouse epithelial cell culture medium for 24 h to 5 days, with 15,000 cells/well in 96-well plates or 50,000 cells/well in 6-well plates.

### 2.3. Sample Collection and Tissue Processing

Broncho-alveolar lavage (BAL) fluid and cells were recovered from lungs by washing twice with 500 µL of cold sterile PBS. Cells were recovered by centrifugation and enumerated by manual counting using a hemocytometer, then cytocentrifuged and stained with Hema-3 fixative solutions (Thermo Fisher Scientific, Waltham, MA, USA) for differential cell analysis. After BAL collection, the left lobe of lungs was perfused with 10% formalin for 48 h then transferred and stored in 70% ethanol. A section of the right lobe was frozen in liquid nitrogen and stored in −80 °C until further processing. Frozen tissue samples were crushed and equal portions were resuspended in radioimmunoprecipitation assay (RIPA) buffer containing protease inhibitors (Aprotinin, PMSF, Na_3_VO_4_, DTT) for protein, or Trizol (Thermo Fisher Scientific, Waltham, MA, USA) for RNA extraction. Protein samples were processed using a homogenizer and RNA samples purified by phenol-chloroform extraction. Samples were stored in −80 °C until further analysis.

### 2.4. Histology, Immunohistochemistry and Chromogenic In Situ Hybridization (CISH)

Following formalin fixation, the left lobe of lungs was dissected into three sections per lung, embedded in paraffin and stored at rt for further analysis. Sections of 3 µm were cut and stained with hematoxylin and eosin (H&E) to assess lung pathology and periodic acid-Schiff (PAS) to assess mucous-producing goblet cells. Immunohistochemistry for Ki67 to assess cell proliferation and alpha smooth muscle actin (αSMA) was also performed. Slide images were captured using Zeiss Axio Imager 2, or scanned using Aperio ImageScope. Quantification of Ki67^+^ cells was performed on ImageJ, and quantification of αSMA^+^ staining on QuPath. Chromogenic in situ hybridization (CISH) for mouse RELMα, YM-1 and CD68 mRNA was performed using the RNAscope^®^ 2.5 Duplex Assay Kit (Advanced Cell Diagnostics, Newark, CA, USA) and stained the BOND lx automated staining instrument (Leica). Formalin-fixed paraffin-embedded lung tissue sections were pretreated with heat and protease prior to hybridization with the target oligo probes. Specific RNA staining signal was identified as punctate dots.

### 2.5. Flow Cytometry

Whole lung mononuclear cell suspensions were generated from a section of the right lobe by mechanical mincing and collagenase digestion. Tissues were degraded with Collagenase I and DNase I for 2 h on an incubated shaker at 37 °C. Debris were removed by passage through 45 µm mesh filter and cells were resuspended in 1 × PBS/0.3% bovine serum albumin (BSA). Cells were stimulated with phorbol 12-myristate 13-acetate (PMA) and ionomycin for 4 h prior to flow cytometric analysis. Forward scatter and side scatter parameters and Zombie-Aqua dye (BioLegend, San Diego, CA, USA) were used to define the live cell gate. Cells were stained with Zombie-Aqua dye for 20 min at room temp (RT) prior to surface staining with fluorophore-conjugated antibodies. Cells were surface-stained for 30 min at 4 °C with antibodies for CD3, CD4, CD38, and CD45 (BD Biosciences, San Jose, CA, USA). Following surface staining, cells were fixed with BD Cytofix/CytoPerm (BD Biosciences, San Jose, CA, USA) for 20 min at 4 °C and then washed with BD 1 × Perm/Wash (BD Biosciences, San Jose, CA, USA) prior to intracellular staining for 30 min at 4 °C with antibodies for CD206 and IFNγ (BD Biosciences, San Jose, CA, USA). Cells were then washed and resuspended in 1 × PBS/0.3% BSA for analysis on the BD LSR II flow cytometer.

### 2.6. Western Blot

BAL fluid or culture supernatants were loaded onto 15% SDS-PAGE gels and separated by electrophoresis at 120 V for 1 h, then transferred to nitrocellulose membranes at 400 mA for 1 h. Blots were blocked using Odyssey Blocking Buffer (LI-COR Biosciences, Lincoln, NE, USA) for 1 h at RT, then probed using rabbit anti-RELMα polyclonal antibody (PeproTech, Rocky Hill, NJ, USA) and goat anti-β-actin polyclonal antibody (Santa Cruz Biotechnology, Dallas, TX, USA) overnight at 4 °C. The following day blots were washed with 1 × Tris-buffered saline (TBS) + 0.15% Tween20, and anti-rabbit and anti-goat secondary antibodies (LI-COR Biosciences, Lincoln, NE, USA) were added to blots for 45 min at RT. After subsequent washes, blots were imaged using an Odyssey LI-COR Imaging System.

### 2.7. Reverse Transcription Polymerase Chain Reaction (RT-PCR) / NanoString

Lung RNA purified by phenol-chloroform extraction were reverse transcribed into cDNA and stored at −80 °C. cDNA was analyzed by Taqman^®^ assay (Applied Biosystems, Foster City, CA, USA) using pre-determined assay reagents and specific probes for RELMα and 18S. Gene expression was also measured by NanoString Technologies (Seattle, WA, USA). Analysis of raw mRNA counts was performed using nSolver^TM^ Analysis Software v4.0. Gene expression levels were normalized to housekeeping genes *Actb* and *Pgk1*. Raw Nanostring data are shown, where counts below 10 are considered below the limit of detection.

### 2.8. Enzyme-Linked Immunosorbent Assay (ELISA)

BAL fluid and serum samples from mice were analyzed using commercially available DuoSet ELISA kits from R&D Systems (Minneapolis, MN, USA) for YM-1 and eotaxin-2. For measuring RELMα by ELISA, an in-house sandwich ELISA was developed using an anti-mouse RELMα polyclonal antibody and a biotinylated anti-mouse RELMα polyclonal antibody (PeproTech, Rocky Hill, NJ, USA).

### 2.9. Statistical Analysis

Statistical analyses were carried out using GraphPad Prism 7. The Student’s t test, one-way or two-way analysis of variance was used to determine significant differences between sample groups, where **p* < 0.05, ***p* < 0.01, ****p* < 0.001, *****p* < 0.0001.

## 3. Results

### 3.1. RELMα is Induced Upon Overexpression of OSM in Lungs of C57Bl/6 Mice and is Highly Expressed in Airway Epithelial Cells

To examine the in vivo regulation of RELMα, C57Bl/6 were endo-tracheally administered with PBS, the empty control vector AdDel70, or AdOSM to induce transient overexpression of mOSM as previous [13,36,37] in the lungs for seven or 14 days. Lung tissues were assessed for RELMα mRNA expression, and BAL fluid and serum were analyzed for RELMα protein. In AdDel70-treated C57Bl/6 mice, RELMα mRNA was detectable at low levels, which was significantly up-regulated upon treatment with AdOSM at day 7 (Figure 1A, left panel). At the protein expression level (as assessed by ELISA), RELMα was detectable at basal levels (~100 ng/mL) in BAL fluid of naïve and AdDel70-treated C57Bl/6 mice, and 7 and 14 days following overexpression of OSM, RELMα protein detected was markedly induced ~180-fold to up to approximately 12 µg/mL at day 7, and afterwards decreased to approximately 4 µg/mL by day 14 (Figure 1A, middle panel). RELMα was present at 150–200 ng/mL in serum of control animals and elevated upon AdOSM infection at both days 7 and 14 in C57Bl/6 mice (Figure 1A, right panel).

Given that previous studies have shown that RELMα can be expressed in other cell types in addition to AA/M2 macrophages [26,38], we sought to determine the cell sources of RELMα following AdOSM treatment. Lung tissue sections from C57Bl/6 wild-type were analyzed for mRNA by chromogenic in situ hybridization (CISH) to identify mRNA signals (Figure 1B and Figure 2). In naïve and AdDel70-treated lungs, RELMα was expressed in few cells of the airway epithelium. Upon overexpression of OSM for seven days, RELMα was highly expressed in columnar airway epithelial cells and was also expressed in mononuclear cells throughout the lung parenchyma (Figure 1B). Lung tissue sections (Day 7 after treatment) were further analyzed by co-stain for RELMα and CD68 (a marker of macrophages) in Figure 2. There was no detectable CISH staining for RELMα in RELMα^–^-deficient lungs as expected (Figure 2A lower panels). CD68^+^ RELMα^−^ cells were found throughout the parenchyma across all treatments. In the lung parenchyma of AdOSM-treated wild-type mice, CD68^+^/RELMα^+^ macrophages were also observed, as demonstrated by co-localization of both stains (Figure 2B, left panel), however many cells in the parenchyma did not co-localize RELMα and CD68, while RELMα was strongly positive in airway epithelial cells (upper portion of 2A upper right panel). To determine if airway epithelial cells can respond to OSM directly, we assessed cell signaling response of primary mouse airway epithelial cells to OSM and other cytokines (Figure 2C). These cultures responded robustly with phospho-STAT3 elevation to OSM, to a low extent to IL-6, but not other gp130 cytokines LIF or IL-31, or to IL-4. Full blots are shown in Appendix A.

### 3.2. IL-6 and STAT3 are not Required for Airway Epithelial Cell Responses to OSM

As a comparator of OSM activity to the prototypical gp130 cytokine IL-6, AdIL-6 was administered to C57Bl/6 mice to induce transient overexpression of mouse IL-6 for two and seven days and compared to mice treated in parallel with AdOSM. RELMα in BAL fluid quantified by ELISA (Figure 3A, left panel) showed a similar trend in that AdIL-6 induced RELMα but to markedly lower levels than AdOSM (2 vs 6 µg/mL at Day 7), and where RELMα levels at day 2 appeared elevated but not statistically different from controls. Western blots (right panels of Figure 3A–C) of whole lung extracts showed that a full length, single band was detected at the expected molecular weight (~10 kDa), and reflected the observations found in BAL. An example of full blots probed for RELMα with recombinant RELMα as a comparator for size and amount is depicted in Appendix A.

To determine if IL-6 is required for AdOSM-induced RELMα expression, wild-type C57Bl/6 and IL-6^–/–^ mice were administered AdDel70 or AdOSM, and whole lung extracts were assessed for RELMα by ELISA (Figure 3B, left panels). In these assays of whole lung extracts, RELMα was low/non-detectable in AdDel70-treated lungs, and elevated significantly by AdOSM in wt or IL-6^-/-^ mice (Figure 3A,B), suggesting the IL-6 was not required for total lung RELMα protein load. Since skewing of macrophages populations toward AA/M2 phenotypes involves signals induced by IL-4/IL-13 and STAT6 conical signaling, we assessed the level of RELMα in STAT6^-/-^ mice. Total lung RELMα levels induced by AdOSM were not significantly reduced in STAT6^-/-^ mice (Figure 3C). CISH staining for RELMα mRNA in lung tissue sections of both IL-6^-/-^ and STAT6^-/-^ animals are shown in Figure 4. There was no detectable staining in sections processed without a specific probe for RELMα (Appendix A). In either strain of IL-6^-/-^ or STAT6^-/-^ mouse lungs, AdOSM induced RELMα mRNA in the lung airway epithelial cells, suggesting no absolute requirement of either of these pathways for OSM induction of these cells. There was minimal change qualitatively in IL-6^-/-^ mouse lung parenchymal RELMα mRNA signals (Figure 4A,B), but a clear qualitative decrease in RELMα mRNA signals in the parenchymal cells in STAT6^-/-^ mouse lung sections (Figure 4C,D).

### 3.3. YM-1 Is Induced by AdOSM

Since YM-1 is another well-recognized secreted product and marker of AA/M2 macrophages, we also assessed the expression of YM-1 in the AdOSM model. Figure 5A (left panel) shows that AdOSM but not AdIL-6 elevated YM-1 protein was found in whole lung extracts, and this was significantly reduced in IL-6^-/-^ whole lung extracts (Figure 5A middle panel) but not in STAT6^-/-^ mice (right panel). To determine which cells express YM-1 in this system, we completed CISH analysis staining for YM-1 mRNA (green-blue) and OSM mRNA (red) on tissue sections of Addel70 and AdOSM treated mouse lungs (Figure 5B,C). YM-1 mRNA+ cells were markedly increased in the parenchyma of AdOSM treated animals compared to Addel70, and primarily localized to mononuclear cells. Interestingly, YM-1 mRNA signal was also evident in the airway epithelial cells of AdOSM treated mice, although not as prominent as RELMα (Figure 1 and Figure 2). Strong OSM mRNA signals were observed in some of the airway epithelial cells, reflecting the vector-encoded OSM mRNA and the well-established cell tropism of Adenovirus constructs for these cells. There was no evidence of staining in sections using the detection systems without specific probe for YM-1 or OSM (Appendix A), either in the parenchyma or the airway epithelium (both regions are shown in the control section).

### 3.4. RELMα Supports Maximal CD206^+^ M2 Macrophage Numbers Induced by AdOSM

Although RELMα has been implicated in several different murine models of allergic inflammation, pulmonary fibrosis, and helminth infection, its role in type 2 inflammation in the lung is not fully understood. To investigate functions of RELMα in OSM-mediated lung inflammation, wild-type and RELMα^–/–^ mice were endo-tracheally administered PBS, AdDel70, or AdOSM for 7 days, and BAL fluid and lung tissue were examined. RELMα mRNA in naive or AdOSM-induced lungs was non-detectable in RELMα^–/–^ mice, as assessed by Nanostring analysis of RNA extracted from whole lung tissue (Figure 6A, left panel), nor was RELMα protein detectable by ELISA of the BAL fluid (Figure 6B) or as assessed in Western blot analysis of whole lung homogenates of RELMα^–/–^ mice (Figure 6C). OSM mRNA in whole lung and protein in BAL fluid were also measured to compare overexpression of the cytokine in AdOSM-treated animals. The absence of RELMα did not affect the overexpression of Adenovirus-encoded OSM mRNA or protein in whole lung or BAL fluid as assessed by Nanostring or ELISA (Figure 6A,B, right panels). Analysis of Eotaxin-2, previously shown to be induced by AdOSM (22), and YM-1 protein in BAL showed robust elevation by AdOSM and induction was not altered in RELMα^-/-^ mice (Figure 6B lower panels). Consistent with previous findings [22], AdOSM-treated wild-type mice demonstrated accumulation of eosinophils, lymphocytes, and neutrophils in BAL fluid in comparison to control animals. RELMα-deficiency did not appreciably affect the accumulation of these inflammatory cell populations detected at day 7 in BAL (Figure 7A).

We next examined whether RELMα would affect the accumulation of AA/M2 macrophages in lungs treated with AdOSM by flow cytometry (Figure 7B). Macrophages were classified as CD45^+^ (hematopoietic cell marker), F4/80^+^ (macrophage marker), and either CD206^+^ (AA/M2 macrophage marker) or CD38^+^ (M1 macrophage marker). Similar to previous observations [22], AdOSM induced the accumulation of CD206^+^ CD38^–^ AA/M2 macrophages in the lung at day 7, and CD38^+^ CD206^–^ M1 macrophages also accumulated in the lung (Figure 7B). In the absence of RELMα, there was approximately 60% reduction in total numbers of CD206^+^ AA/M2 macrophages in the AdOSM group in comparison to wild-type counterparts, whereas there was no significant effect of RELMα-deficiency on CD38^+^ M1 macrophage detection. Th1 cells were also identified through flow cytometry as CD45^+^, CD3^+^ (T cell marker), CD4^+^ (T helper cell marker), and IFNγ^+^ (Th1 cell marker). At day 7, AdOSM markedly induced the accumulation of IFNγ-producing Th1 cells in the lungs of both wild-type and RELMα^–/–^ mice, with 40% fewer total IFNγ^+^ Th1 cells in knockout animals in comparison to wild-types (Figure 7C). Thus, OSM-induced lung inflammation and accumulation of AA/M2 and Th1 cells may be regulated in part through RELMα.

### 3.5. RELMα-Deficiency Does Not Alter Th2-Associated Cytokine Elevation but Reduces Arginase1 and Matrix Remodelling Gene Induction

We did not observe significant changes of Th2-associated inflammatory cytokine mRNA expression as a result of RELMα-deficiency (Figure 8A). AdOSM up-regulated similar levels of IL-4, IL-5, IL-6, and IL-33 mRNA after 7 days in both wildtype and RELMα^–/–^ mice (Figure 8A,B). In correlation with the IFNγ^+^ Th1 cell population, as assessed by flow cytometry, IFNγ mRNA in total lung was upregulated, albeit to a low extent (2-fold), by AdOSM treatment in wild-type animals, but in RELMα^–/–^ mice its gene expression was similar across all treatment groups (Figure 8A). In our previous work [22] and here, we could not detect IFNγ protein in BAL fluid. Consistent with our previous findings [22], AdOSM induced Arg1 mRNA, and its expression was significantly reduced in the absence of RELMα (Figure 8B). Together with flow cytometry of CD206^+^ macrophages (Figure 7B), this suggests a potential role for RELMα in maximizing the accumulation of AA/M2 macrophages.

We have previously demonstrated that pulmonary overexpression of OSM can induce ECM deposition in C57Bl/6 mice [13,36]. We therefore examined the role of RELMα in OSM-induced ECM accumulation by analysis of gene expression. Consistent with our previous findings [13], mRNAs for proteins implicated in ECM remodeling, COL1A1, COL3A1, MMP13, and TIMP1, were up-regulated in lungs of AdOSM-treated wild-type mice at day 7 (Figure 8C). Similar trends were observed in RELMα^–/–^ mice. However the expression of these genes was significantly reduced in the absence of RELMα, suggesting that ECM accumulation induced by OSM may be regulated in part through RELMα.

### 3.6. RELMα-Deficiency Reduces Cell Proliferation and Parenchymal αSMA Accumulation

Lung tissue sections were examined by histological analyses to determine the effects of RELMα deficiency on cell proliferation, goblet cell hyperplasia, and accumulation of αSMA. Formalin-fixed, paraffin-embedded lung tissue sections stained with H&E showed thickening of the airway epithelium in AdOSM-treated wild-type mouse lungs, which was reduced in RELMα-deficient animals (Figure 9A). These sections were stained with the cell proliferation marker Ki67 to determine whether airway thickness was due to increased proliferating epithelial cells. However, there did not appear to be obvious differences in Ki67^+^ cells in the epithelium between wild-type and RELMα^–/–^ mice at day 7 (Figure 9B). Quantification of Ki67^+^ cells in these sections did show that AdOSM-treated lungs had a greater proportion of Ki67^+^ cells around the airways and throughout the lung parenchyma, whereas in the absence of RELMα there was a significant reduction in proliferating cells (Figure 9E). PAS staining, an indicator of mucous-producing goblet cells, indicated increased numbers of goblet cells in the epithelium of AdOSM-treated mouse lungs, however we did not observe striking differences between wild-type and RELMα^–/–^ animals (Figure 9C). Lung tissues stained for αSMA (Figure 9D) and analysis of entire lung sections, excluding major airways and blood vessels, revealed that AdOSM induced the accumulation of αSMA^+^ cells in the lung parenchyma of wild-type mice (Figure 9F). In RELMα^-/-^ mice treated with AdOSM, there was significantly less αSMA^+^ staining.

## 4. Discussion

Our results demonstrate that the overexpression of OSM in mouse lungs strongly induces mRNA expression of RELMα in columnar airway epithelial cells (columnar-AEC) and RELMα protein detected in lungs of the C57Bl/6 strain of mice. AEC responded directly to OSM in vitro, and OSM-induced RELMα in columnar-AEC did not require STAT6 or IL-6 in vivo. OSM also induced YM-1 in whole lung, and YM-1 mRNA in columnar-AEC. This is the first report of gp130 cytokine regulation of RELMα independently of typical Th2 cytokine environments, and of YM-1 expressed by AEC. Furthermore, RELMα-deficiencies in C57Bl/6 mice did not affect the expression of adenovirus-encoded OSM, nor of mRNA expression of Th2-associated mediators such as IL-4, IL-5, IL-6, and eotaxin-2, nor eosinophil accumulation in BAL fluid. However, the absence of RELMα in deficient mice resulted in less accumulation of AA/M2 macrophages in comparison to wild-type counterparts, less up-regulation of ECM remodeling genes, and less parenchymal αSMA^+^ staining in knockout animals treated with AdOSM, characteristics of altered tissue repair and wound healing phenotypes. This suggests an OSM-RELMα pathway in AEC integrated into tissue healing/repair responses.

Among the gp130 cytokines, the receptor chains for the OSM receptor complex (OSMRβ and gp130) are widely expressed on connective tissue cells, and can engage several signaling pathways including JAK/STAT, MAPK, and PI3K/Akt [15,17]. Based on in vitro studies [18,39], OSM is more active than other gp130 cytokines, such as IL-6, LIF, or IL-31, and robustly regulates the expression of signaling intermediates, including STAT3 and STAT1, as well as genes including IL-4Rα, IL-6, and ECM genes MMP1 and TIMP1 in connective tissue cells. A similar trend of OSM potency was also observed here in vivo (comparing RELMα and YM-1 response to OSM verses IL-6 over-expression Figure 3A and Figure 5A) and in vitro (comparing pSTAT3 activation, Figure 2C). In lung inflammation with OSM detectable in human disease conditions, OSM may thus engage AEC responses in inflammatory cascades, and although we have shown that human BEAS2B AEC cells respond to OSM in vitro with pSTAT3 activation [13], other endpoints in human airway EC need to be fully explored. OSM is produced by macrophages upon inflammatory stimuli and thus may participate in AEC regulation in TH1-skewed inflammatory environments as well as TH2-skewed local tissue environments.

The use of CISH to determine cellular localization of mRNA signals is useful in examining complex inflammatory models in vivo. Here we show that RELMα was expressed in some CD68+ macrophages as expected (presumably AA/M2 macrophages) but in addition CD68- cells in the parenchyma (Figure 2). Others have shown that RELMα can be expressed by alveolar type II epithelial cells using RELMα promoter reporter mice [40]. Thus, alveolar type II epithelial cells may also be responsive to OSM with RELMα expression (we have published that these cells express IL-33 in response to OSM in vitro and in vitro). Our results showing that parenchymal RELMα is lost in STAT6^-/-^ mice suggest that IL-4/13/STAT6 canonical signaling is required for RELMα induction in lung macrophages as previously described [41], but in addition type II alveolar epithelial cells (Figure 4C,D). However, further work is required to explore the STAT6 requirement of Alveolar epithelial cells in this system in vivo. The lack of change in total lung extract RELMα levels may reflect a predominant contribution of airway epithelial cells to total lung RELMα protein load. Parenchymal mRNA signal induction for RELMα was not affected in IL-6^-/-^ mice (Figure 4A,B), suggesting a requirement of STAT6 but not IL-6 in alveolar type II epithelial cell or alveolar macrophage expression of RELMα.

In addition to AEC, YM-1 mRNA was elevated in mononuclear cells in the parenchyma (Figure 5B,C) consistent with AAM/M2 expression of this gene product. IL-6^-/-^ mice showed a decrease in YM-1 total protein (Figure 5A), which is consistent with the ablation of accumulation of Arg1+ AAM/M2 cells observed previously in IL-6^-/-^ mice [22]. However, total YM1 was not appreciably ablated in STAT6^-/-^ mice (Figure 5A). YM-1 mRNA was not expressed by alveolar type II cells and thus we suggest that airway epithelial cells contribute significantly to total YM-1 protein levels in this model, supported by the expression of YM-1 mRNA in airway epithelial cells (Figure 5B,C).

RELMα has been implicated in several different murine models of lung inflammation, including parasitic infection and pulmonary fibrosis. In a model of pulmonary inflammation, RELMα^–/–^ mice challenged with the parasite *Schistosoma mansoni* (*Sm*) eggs presented elevated Th2 cytokine levels including IL-4, IL-5, and IL-13 in comparison to wild-type counterparts [26]. However, in a different model of pulmonary fibrosis where RELMα^–/–^ mice were treated with bleomycin, these mice exhibited decreased IL-4 expression and less fibrosis in comparison to wild-type mice [33]. In our model of Th2-skewed pulmonary inflammation, Th2 cytokines were not differentially expressed between wild-type and RELMα knockout mice, suggesting the Th2 cell responses were not altered. We did observe lower IFNγ mRNA expression in RELMα^–/–^ mice (Figure 7A), as well as reduced IFNγ-producing CD4^+^ Th1 cells in total lung of knockout mice relative to wild-type mice (Figure 6C). However, in this and previous studies [22], we cannot detect IFNγ or IL-12 in BAL fluid, and the mRNA for IFNγ is low and minimally induced (two-fold) by AdOSM in total lung (Figure 8A). That being noted, it appears that RELMα can regulate activated Th1 cells although it is not clear if this is a direct or indirect action. Taken together, the function of RELMα in regulating Th1 and Th2 responses appears dependent on the specific model of lung inflammation.

In a model of type 2 immunity during nematode infection, studies have shown that Ym1 can induce epithelial cell expression of RELMα [42]. Ym1 is another AA/M2 macrophage- associated protein elevated in Th2 inflammation and AdOSM induces Ym1 in our model system in vivo. Whether OSM directly induces RELMα in columnar airway epithelial cells (OSM can directly induce AEC STAT3 signaling), or indirectly through Ym1 (or possibly both pathways), requires further study, since we were not able to detect RELMα production in mouse airway epithelial cells in liquid culture in vitro (data not shown). This may reflect the environment present in vivo (additional factors and/or AEC phenotype) is not recapitulated in in vitro liquid cell culture. It would also be of interest to investigate the expression and potential requirement in inflammation of OSM in *N. brasiliensis* infections, in particular in the lung inflammatory phase, since OSM may participate in the RELMα and Ym1 induction in such nematode infections.

In our system of OSM-induced lung inflammation, although there were no significant differences in Th2 cytokine expression between wild-type and RELMα^–/–^ mice, we did observe a lower fibrotic response in knockout mice treated with AdOSM. Similar to the bleomycin model of pulmonary fibrosis [33], we observed reduced type I collagen lung mRNA expression (Figure 8C), correlating with markedly less αSMA^+^ staining in the lung tissue of RELMα^–/–^ mice relative to wild-type mice (Figure 9F), suggesting that knockout mice were partially protected from this particular fibrotic phenotype, and that RELMα may participate in myofibroblast accumulation. Several studies have suggested a role for AA/M2 macrophages in the exacerbation of pulmonary fibrosis and other models of tissue repair and wound healing [43,44]. Here, we show that AdOSM induced the accumulation of CD206^+^ M2 macrophages, whereas in the absence of RELMα, there was significantly less accumulation of these macrophages in the lung (Figure 7B). This trend was also associated with reduced lung mRNA expression of ECM remodeling proteins COL1A1, COL3A1, MMP13, and TIMP1 in knockout animals (Figure 8C). Taken together, RELMα may function to maintain M2 macrophage numbers in the lung, which may then contribute to myofibroblast differentiation and ultimately lead to matrix deposition and lung fibrosis.

How RELMα acts on various cells mechanistically is not clear. A receptor for RELMα remains to be fully identified, although some have suggested that RELMα could bind to intracellular Bruton’s tyrosine kinase (BTK) [26,45]. While the interaction between the two is unclear since RELMα is secreted whereas BTK is cytoplasmic, activation of the BTK pathway can lead to myeloid cell chemotaxis, and in a different study, down-regulate Th2 cytokine production in CD4^+^ T cells [26,45]. Others have proposed that RELMα could induce myofibroblast differentiation through the induction of αSMA in fibroblasts [46], however it is unclear how RELMα binds or activates these fibroblasts. RELMα also induces myofibroblast transition of mouse adipocytes and this may play a role in dermal fibrosis [47].

RELMα protein was present basally in serum of naïve mice at approximately 200 ng/mL (Figure 1A), however it is unclear whether RELMα plays a role in systemic circulation in this model. Since the levels of RELMα in BAL fluid can be induced to as high as 12 µg/mL, it is unlikely that the source of RELMα in lung comes from circulating immune cells, but rather the increase in RELMα levels in serum of AdOSM-treated C57Bl/6 mice is due to spillover or absorption from BAL fluid. Recently, RELMα has been shown implicated in whole-body metabolism in vivo. Kumamoto et al. demonstrated that CD301b^+^ mononuclear phagocytes (MNP) in white adipose tissue were a major source of RELMα [28]. Depletion of these cells disrupted metabolic homeostasis, resulting in significant weight loss and reduced blood glucose levels, and reconstitution of RELMα by intra-peritoneal administration was able to reestablish glucose homeostasis. Whether the increases of serum RELMα in our system of AdOSM overexpression in C57Bl/6 mice has effects on glucose homeostasis would require further study.

To date, the human homolog for mouse RELMα has yet to be identified. Some have suggested that human RELMβ is a functional homologue to mouse RELMα [48]. One study demonstrated that human RELMβ was present in epithelial cells and alveolar macrophages of human lung tissue from IPF patients [49], consistent with murine models showing RELMα expression in the same cell types [23]. In another study, RELMβ was found at significantly higher levels in bronchial biopsies of patients with asthma compared to healthy individuals and, as the authors suggest, RELMβ may play a significant role in airway remodeling mechanisms in human asthma [50]. Further investigation is required to determine whether OSM regulates RELMβ in mouse or human lung stromal/epithelial cells.

## 5. Conclusions

In summary, we have shown that the overexpression of OSM induces a robust expression of RELMα in mouse lungs, induces RELMα in airway epithelial cells without the requirement of IL-6 or STAT6 in vivo, and can directly activate airway epithelial cells in vitro. RELMα is required in this system for maximal induction of ECM modulating genes, AA/M2 macrophage accumulation and αSMA, but not inflammatory cytokines. These interactions are depicted schematically in the accompanying Scheme 1. Since OSM is elevated in chronic lung diseases in humans, determining the function of the RELM family proteins in OSM-mediated lung inflammation may serve to clarify mechanisms of pathogenesis and/or homeostasis in such conditions.

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
