# Peer review of "RELMα Is Induced in Airway Epithelial Cells by Oncostatin M without Requirement of STAT6 or IL-6 in Mouse Lungs In Vivo"

_cells, 2020, doi:10.3390/cells9061338_

Round 1
Reviewer 1 Report
Authors describe their studies assessing the induction of RELMa by OSM in the mouse lung, including requirement for STAT6 and IL-6. Authors also explore YM-1 regulation. Overall, the findings are of interest and extend our understanding of the role of RELMa in the lung inflammatory response. However, several concerns need to be addressed, as detailed below:
1 - Authors do not present controls for various figures that limit the ability of readers to assess the validity of the described results. This is an issue throughout the figures, including lack of negative control for YM-1 or OSM probes or a magnified image for the control in Figure 5. This is particularly relevant given that authors are claiming this to be the first instance showing YM-1 expression in AEC. Chosen images in Figure 5 also do not contain the same approximate airway/alveolar tissue distribution. Similarly, no naive/control staining in the KO animals are provided in Figure 4, no naive staining in Figure 1, etc.
2 - Authors should verify the appropriate statistical tests to use, e.g. is a one-way ANOVA appropriate when considering changes across multiple time-points for different treatments (e.g. Figure 1), and a one-way ANOVA is also not appropriate for Figure 5A studies.
3 - Insufficient details are given regarding the Nanostring expression analyses. Was background subtraction performed and, if so, what count threshold was used for determining what counts to use? Many of the reported counts are very low (IL-4, IL-5, IFNg).
4 - There is a lack of mechanistic insight from the presented results that overall limits enthusiasm for the work. At the very least, additional discussion of potential underlying mechanisms, and limitations of the current study is needed.
5 - Authors should present total macrophage numbers from the differential cell counts described in Figure 7A.
6 - There are several more minor issues that need to be fixed. Figure 1 indicates an N of 4-6, but it is clear that naive animals have only N=3. Also, staining in Figures 2 and 5 are described as green, but the color looks blue in print. Images throughout need to be checked for errors in formatting, e.g. special symbols, etc. in Figure 2C, Figure 5, Figure 6, Figure 9. On page 13, line 361 authors incorrectly reference Figure 9C.
Author Response
We thank the reviewers for their thoughtful critique and have now revised the manuscript as indicated below
Reviewer 1 points 1-6
1 - Authors do not present controls for various figures that limit the ability of readers to assess the validity of the described results. This is an issue throughout the figures, including lack of negative control for YM-1 or OSM probes or a magnified image for the control in Figure 5. This is particularly relevant given that authors are claiming this to be the first instance showing YM-1 expression in AEC. Chosen images in Figure 5 also do not contain the same approximate airway/alveolar tissue distribution. Similarly, no naive/control staining in the KO animals are provided in Figure 4, no naive staining in Figure 1, etc.
Response: We have now included negative controls (all staining procedures except for inclusion of ACD-designed specific probes) for Fig 5 YM-1 and OSM Chromogenic in situ hybridization (CISH) and show this in supplementary figures 3C. As indicated in the text (page 9, line 291) “there was no evidence of staining in sections using the detection systems without specific probe for YM-1 or OSM (Supplementary fig 3C) , either in the parenchyma or the airway epithelium (both regions are shown in the control section)”.
(we did not change chosen images in Fig 5B,C since supplementary Fig 3C shows both parenchyma and airway epithelium)
Control staining (no probe) for IL-6KO or STAT6KO animals in Fig 4 experiments are now provided in Supplementary fig 3A,B, and we indicate in the text (page 8 , line 260 ) that “there was no detectable staining in sections processed without specific probe for RELMα (supplementary fig 3 A,B) “.
Naïve staining for fig 1B is now incorporated (lower panel) and indicated in fig 1 legend
2 - Authors should verify the appropriate statistical tests to use, e.g. is a one-way ANOVA appropriate when considering changes across multiple time-points for different treatments (e.g. Figure 1), and a one-way ANOVA is also not appropriate for Figure 5A studies.
Response: The figure legend for 5A has now been corrected accordingly, as one-way ANOVA was only appropriate for the Figure 5A left panel while two-way ANOVA was completed for the middle and right panels. (The tests were verified for Figure 1).
3 - Insufficient details are given regarding the Nanostring expression analyses. Was background subtraction performed and, if so, what count threshold was used for determining what counts to use? Many of the reported counts are very low (IL-4, IL-5, IFNg).
Response: We have corrected this omission (background nanostring counts were not subtracted) with revised methods text on page 4 line 157 as follows:
“Raw nanostring data are shown, where counts below 10 are considered below the limit of detection”
4 - There is a lack of mechanistic insight from the presented results that overall limits enthusiasm for the work. At the very least, additional discussion of potential underlying mechanisms, and limitations of the current study is needed.
Response: Potential underlying mechanisms and various limitations of the study are incorporated throughout the discussion. We have also now modified the discussion, removing unnecessary text and added a referral to a graphical scheme (as per reviewer 2 suggestion) to help summarize the contents of the manuscript
5 - Authors should present total macrophage numbers from the differential cell counts described in Figure 7A.
Response: Total macrophage counts in BAL now been incorporated into a revised Fig 7A and the legend altered accordingly (line 336)
6 - There are several more minor issues that need to be fixed. Figure 1 indicates an N of 4-6, but it is clear that naive animals have only N=3.
Response: This is now corrected in the text of legend for Fig 1
Also, staining in Figures 2 and 5 are described as green, but the color looks blue in print.
Response: We have now described this as green-blue in the text
Images throughout need to be checked for errors in formatting, e.g. special symbols, etc. in Figure 2C, Figure 5, Figure 6, Figure 9. On page 13, line 361 authors incorrectly reference Figure 9C.
Response: We have now corrected these errors.
Reviewer 2 Report
In their study, Ho and co-workers show that transduction-mediated de novo expression of oncostatin M in mouse lungs resulted in accumulation of M2 macrohpages in this tissue, and an overall Th2-biased immune response. Oncostain M-overexpression was accompanied by induction of RELM-alpha and YY-1, but did not depend on IL-6 and STAT-6. RELM-knockout animals reacted in a similar manner, albeit at lower extent.
Minor issues:
1. In several figures, part of the labeling needs to be corrected.
2. I suggest to summarize the numerous findings of this study concerning the role of the various adaptor/effector molecules and their interdependency in a graphical scheme.
Author Response
reviewer 2, points 1-2
Minor issues:
- In several figures, part of the labeling needs to be corrected.
Response: These have now been corrected as per note to reviewer 1 (item 6)
- I suggest to summarize the numerous findings of this study concerning the role of the various adaptor/effector molecules and their interdependency in a graphical scheme.
Response: We have now submitted a graphical scheme to accompany the manuscript